# Deep Network Architectures as Feature Extractors for Multi-Label Classification of Remote Sensing Images

Marjan Stoimchev [1,2,*], Dragi Kocev [1,2,3] and Sašo Džeroski [1,2]

1   Department of Knowledge Technologies, Jožef Stefan Institute, Jamova cesta 39, 1000 Ljubljana, Slovenia;
    dragi.kocev@ijs.si (D.K.); saso.dzeroski@ijs.si (S.D.)
2   Jožef Stefan International Postgraduate School, 1000 Ljubljana, Slovenia
3   Bias Variance Labs, 1000 Ljubljana, Slovenia
*   Correspondence: marjan.stoimchev@ijs.si

**Abstract:** Data in the form of images are now generated at an unprecedented rate. A case in point is remote sensing images (RSI), now available in large-scale RSI archives, which have attracted a considerable amount of research on image classification within the remote sensing community. The basic task of single-target multi-class image classification considers the case where each image is assigned exactly one label from a predefined finite set of class labels. Recently, however, image annotations have become increasingly complex, with images labeled with several labels (instead of just one). In other words, the goal is to assign multiple semantic categories to an image, based on its high-level context. The corresponding machine learning tasks is called multi-label classification (MLC). The classification of RSI is currently predominantly addressed by deep neural network (DNN) approaches, especially convolutional neural networks (CNNs), which can be utilized as feature extractors as well as end-to-end methods. After only considering single-target classification for a long period, DNNs have recently emerged that address the task of MLC. On the other hand, trees and tree ensembles for MLC have a long tradition and are the best-performing class of MLC methods, but need predefined feature representations to operate on. In this work, we explore different strategies for model training based on the transfer learning paradigm, where we utilize different families of (pre-trained) CNN architectures, such as VGG, EfficientNet, and ResNet. The architectures are trained in an end-to-end manner and used in two different modes of operation, namely, as standalone models that directly perform the MLC task, and as feature extractors. In the latter case, the learned representations are used with tree ensemble methods for MLC, such as random forests and extremely randomized trees. We conduct an extensive experimental analysis of methods over several publicly available RSI datasets and evaluate their effectiveness in terms of standard MLC measures. Of these, ranking-based evaluation measures are most relevant, especially ranking loss. The results show that, for addressing the RSI-MLC task, it is favorable to use lightweight network architectures, such as EfficientNet-B2, which is the best performing end-to-end approach, as well as a feature extractor. Furthermore, in the datasets with a limited number of images, using traditional tree ensembles for MLC can yield better performance compared to end-to-end deep approaches.

**Keywords:** remote sensing; convolutional neural networks; tree ensemble methods; multi-label classification

## 1. Introduction

Remote sensing is the process of detecting and monitoring the physical characteristics of an area by measuring its reflected and emitted radiation at a distance. In the past few years, advances in satellite technology have resulted in large-scale remote sensing image (RSI) archives, which have attracted a considerable amount of research in various application areas. RSI can be used to monitor and predict various environmental phenomena, such as weather and climate change [1], land use and land cover changes at macro scale [2], deforestation [3], wildfires [4,5], and many others.

In machine learning terms, a wide range of applications stemming from RSI are approached using single-label classification, where the goal is to assign a single label/semantic category to an image [6]. However, real-world RSIs are typically complex and present more than a single semantic category within a single image. Hence, single-label classification is often insufficient to fully describe the presence of complex areas, which can carry semantically complex content [7].

To facilitate more realistic representation of the content of RSI, the analysis of RSI should be addressed through the task of multi-label classification (MLC), where a given image can be associated with multiple semantic concepts/labels taken from a predefined set of labels. In this way, the classification problem becomes more challenging as compared to single-label classification, as discussed in a recent large-scale comparative study and analysis of a wide range of MLC methods [8]. This study highlights the two major challenges that can limit the performance of MLC methods: the presence of complex label correlations and high-dimensional label spaces.

These challenges are attracting increasing amounts of attention from researchers focusing on deep learning (DL) methods, specifically on methods capable of automatically learning long-range dependencies (e.g., with the use of self-attention mechanisms in the vision transformer (ViT) base network architectures [9]), and handling the high-dimensional label spaces. Their internal mechanisms and structure, such as the hierarchical design and characteristics in convolutional neural networks (CNNs), with local connectivity and non-linearity, are capable of encoding information exceptionally well. Moreover, their flexible design offers knowledge to be extracted and discriminative representations to be learned from noisy data in an end-to-end manner, and achieves more accurate recognition performance in less-constrained environments as compared with traditional MLC approaches. Furthermore, the recent success of these methods can be associated with their ability to leverage large amounts of labeled data in order to learn meaningful knowledge.

Many of the existing approaches in computer vision try to address the challenges encountered by exploiting proven DL network architectures pre-trained on large-scale and diverse datasets such as ImageNet [10]. This is usually achieved by using the transfer learning paradigm [11], where specific parts of the model are fine-tuned in order to learn new features that generalize better to the new downstream task [12–14]. For example, Wang et al. [15] propose a framework based on a VGG-16 CNN model initialized with weights learned on ImageNet, which is used to extract semantic representations from images and is coupled with a recurrent neural network (RNN) network architecture with long short-term memory (LSTM) units to capture image/label relations and label dependency. Chen et al. [16] propose an end-to-end learning method based on Graph Convolutional Networks (GCN) to capture the label correlations, and a novel re-weighting scheme for creating the label correlation matrix that is used to guide the information propagation among the nodes in the GCN.

In addition, with the increased availability of RSI and the increased research interest in remote sensing applications, many interesting approaches have been proposed at the crossroads of remote sensing and computer vision [7,17–19]. For example, Hua et al. [20] propose a novel approach for MLC from aerial imagery—an attention aware label relational network, comprising a label-wise feature parcel module, an attention region extraction module, and a label relational inference module. Sumbul et al. [21] present a K-Branch CNN that uses a multi-attention strategy for bidirectional LSTM networks, which is specially developed to capture spatial and spectral contents from RS images of local image areas. Wang et al. [22], despite using local attention [21], are also able to maintain global context through global attention pooling, where the combination of both helps in modeling long-range dependencies among multiple objects and captures underlying relationships among multiple labels.

In this work, our main focus is on investigating the potential of several prominent deep learning architectures (variants of VGG [23], ResNet [24] and EfficientNet [25]) as feature extractors for the MLC of RSI as well as end-to-end approaches to MLC of RSI. To this

end, we evaluate the performance of the architectures across seven MLC RSI datasets with different properties in terms of number of images, number of labels, and average number of labels per image (label cardinality). More specifically, we use pre-trained network architectures with weights learned on ImageNet as an initialization procedure. We further perform calibration of the pre-trained models by fine-tuning them on a small set of RSIs and compare the fine-tuned and pre-trained MLC performance. Finally, we use the learned models as feature extractors to describe the RSI, and use the resulting feature vectors to learn tree ensembles such as random forests and extra trees for MLC.

The main contributions of this paper can be summarized as follows:

- We present an experimental analysis of different approaches for MLC of RSI. More precisely, we investigate the performance of several deep learning network architectures by using pre-training and fine-tuning as the main learning strategies for the MLC task.
- We evaluate the effectiveness of the deep models used as end-to-end approaches to MLC, and used as feature extractors that provide feature representations of RSI, as inputs to tree ensemble methods for MLC. Moreover, we investigate which of the network architectures is the most suitable choice in terms of performance.
- We also investigate the performance of the considered methods in terms of the influence of the number of labeled training examples by providing the methods with different fractions of the data.

The remainder of this paper is organized as follows. Section 2 describes the MLC RSI data (Section 2.1) and the machine learning methods (Section 2.2) used to analyze them (deep learning architectures in Section 2.2.1 and tree ensembles in Section 2.2.2). Next, Section 3 gives the experimental questions and the specific experimental setup, including parameter instantiations for the methods (Section 3.1), the evaluation strategy (Section 3.2), and the different evaluation measures used to assess the performance of the models (Section 3.3). Furthermore, Section 4 discusses the results of our experiments, focusing on the outcomes of the different representation learning strategies (Section 4.1), deep architectures (Section 4.2), MLC approaches (Section 4.3), and number of images (Section 4.4). Finally, Section 5 concludes the paper by providing a summary of the presented work and directions for further work.

## 2. Materials and Methods

### 2.1. Datasets

We use seven publicly available MLC RSI datasets to assess the performance of the MLC methods, namely UC Merced (UCM) Land Use, AID Multilabel, Ankara HIS archive, DFC-15 Multilabel, MLRSNet, and two variants of the BigEarthNet dataset based on two Corine Land Cover (CLC) nomenclatures (Available at https://land.copernicus.eu/user-corner/technical-library/corine-land-cover-nomenclature-guidelines/html (accessed on 4 January 2023 )), CLC with 43 labels (BigEarthNet-43), and CLC with 19 labels (BigEarthNet-19). When analysing the datasets that have hyperspectral RSI (meaning that they have several spectral bands), such as Ankara and BigEarthNet, we only use the RGB spectral band to train the models.

A summary of the properties of the seven datasets is given in Table 1. We can see that the selected datasets are diverse along several lines: number of images, locations, number of labels, image resolution, and number of labels per example image (label cardinality). This means that the trained predictive models are trained and evaluated in a challenging environments. The selection of typical images from the different datasets with their corresponding class labels is given in Figure 1.

**Table 1.** Description of the used RSI multi-label datasets. $|\mathcal{L}|$ denotes the number of possible labels; *Card* denotes label cardinality (i.e., average number of labels per image); *Dens* denotes label density (average proportion of images labeled with a given label); *N* is the number of images in the dataset, of which $N_{train}$ are in the train and $N_{test}$ in the test datasets; and $w \times h$ is the dimension of the images (in pixels).

| Dataset | Image Type | $|\mathcal{L}|$ | *Card* | *Dens* | *N* | $N_{train}$ | $N_{test}$ | $w \times h$ |
|---|---|---|---|---|---|---|---|---|
| Ankara | Hyperspectral/Aerial RGB | 29 | 9.120 | 0.536 | 216 | 171 | 45 | $64 \times 64$ |
| UC Merced Land Use | Aerial RGB | 17 | 3.334 | 0.476 | 2100 | 1667 | 433 | $256 \times 256$ |
| AID Multilabel | Aerial RGB | 17 | 5.152 | 0.468 | 3000 | 2400 | 600 | $600 \times 600$ |
| DFC-15 Multilabel | Aerial RGB | 8 | 2.795 | 0.465 | 3341 | 2672 | 669 | $600 \times 600$ |
| MLRSNet | Aerial RGB | 60 | 5.770 | 0.144 | 109,151 | 87,325 | 21,826 | $256 \times 256$ |
| BigEarthNet | Hyperspectral/Aerial RGB | 19 | 2.900 | 0.263 | 590,326 | 472,245 | 118,081 | $256 \times 256$ |
| BigEarthNet | Hyperspectral/Aerial RGB | 43 | 2.965 | 0.247 | 590,326 | 472,245 | 118,081 | $256 \times 256$ |

| Ankara | UCM | DFC-15 | AID | MLRSNet | BigEarthNet-19 | BigEarthNet-43 |
|---|---|---|---|---|---|---|
| Bare Soil, Crop (Type-A), Crop (Type-B), Unpaved Road, Grass (Type-A) | bare-soil, buildings, cars, pavement, tanks | impervious, vegetation, building, tree | bare-soil, buildings, cars, court, pavement, trees | bare soil, buildings, grass, trail, wind turbine | Urban fabric, Industrial or commercial units, Inland waters | Discontinuous urban fabric, Industrial or commercial units, Water courses |
| Grass Covered Soil, Bare Soil, Crop (Type-D), Asphalt Pavement, Grass (Type-A) | buildings, pavement, sand, tanks, trees | impervious, vegetation, building | bare-soil, buildings, cars, grass, pavement, tanks, trees | buildings, field, terrace, trail, trees | Arable land, Agro-forestry areas | Non-irrigated arable land, Agro-forestry areas |

**Figure 1.** An illustration of the diversity of images from the different RSI datasets with the corresponding class labels.

### 2.1.1. UC Merced Land Use

The UC Merced data set contains 2100 images grouped into 21 broad categories at the scene level. There are a total of 100 images per category, with the size of $256 \times 256$ and a spatial resolution of 0.3 m. The initial version of this dataset was for single-label classification purposes [26]. Later, Chaudhuri et al. [27] relabeled the images with multiple labels. The total number of distinct object-level labels is 17: airplane, bare soil, buildings, chaparral, court, dock, field, grass, mobile home, pavement, sand, sea, ship, tanks, trees, and water. Each image is annotated with one or more (maximum 7) labels at the object level, containing 3.3 object-level labels per image on average.

### 2.1.2. AID Multilabel

The initial single-label AID dataset [28] was relabeled with multiple labels per image and became the AID multi-label dataset [20]. It is also a more challenging dataset than the U Merced dataset. It contains 3000 aerial images from 30 categories with manually assigned multiple-object labels. The resolution of the images is $600 \times 600$ pixels, where each image has 5.5 object-level labels on average (maximum 11). The spatial resolution varies from 0.3 to 8 m.

### 2.1.3. Ankara HIS Archive

This is a small hyperspectral dataset containing 216 image tiles with a size of $63 \times 63$ pixels [29]. The image patches are obtained by fragmenting large hyperspectral

images, acquired by the NASA EO-1 satellite's Hyperion sensor from the area surrounding the city of Ankara in Turkey. Each image is associated with multiple object-level labels (land-cover-classes) and a single land-use scene-level label where the ground resolution is 30 m. There are 9 object-level labels per image on average, and a maximum of 17. It contains 119 channels of hyperspectral images and corresponding three-channel (RGB) images. We only use the RGB channels.

### 2.1.4. DFC-15 Multilabel

The DFC-15 Multilabel dataset [30] is built from a semantic segmentation dataset (DFC15), first used in the 2015 IEEE GRSS data fusion contest. This dataset is acquired over Zeebrugge with an airborne sensor (300 m off the ground). There are a total of 7 tiles, where each of the tiles is 10,000 × 10,000 pixels with a spatial resolution of 5 cm. The images are assigned pixel-level labels, where each pixel is categorized into 8 object classes: impervious, water, clutter, vegetation, building, tree, boat, and car. The final dataset contains 3342 image patches with 600 × 600 image resolution, where each image is associated with image-level multi-labels.

### 2.1.5. MLRSNet

The MLRSNet dataset is an RSI dataset containing optical satellite images with high spatial resolution [31]. It contains 109,161 RSI annotated with 60 predefined class labels, where the number of labels per image varies from 1 to 13. The images have a fixed resolution of 256 × 256 pixels, where the pixel resolution varies from ∼10 m to 0.1 m. This dataset can be used for a wide range of learning tasks, such as multi-label classification, multi-class classification, multi-label image retrieval, and image segmentation. There are 10 images that do not have labels assigned, as shown in Figure 2; therefore, we exclude these images from the experiments. The final dataset contains 109,151 images.

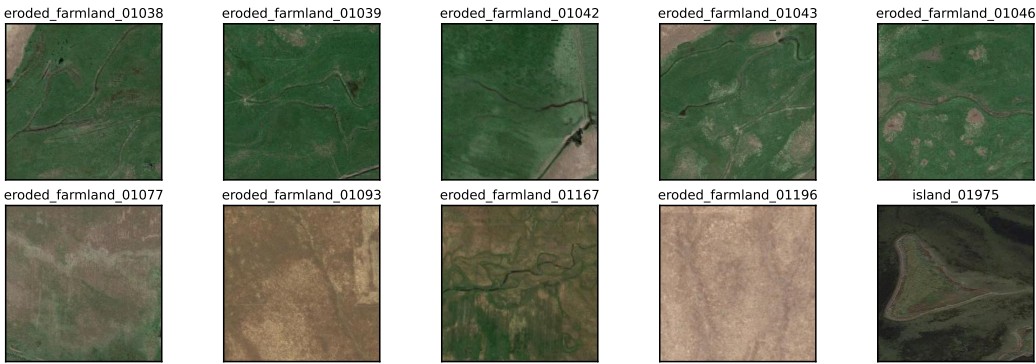

**Figure 2.** Images from the MLRSNet dataset that do not have labels assigned.

### 2.1.6. The BigEarthNet Archive

BigEarthNet [17] was constructed by the Remote Sensing image Analysis (RSiM) Group and the Database Systems and Information Management (DIMA) Group at the Technische Universität Berlin (TU Berlin). It is the largest dataset of image patches annotated with multiple labels available to date. There are 590,326 such patches for which Sentinel-2/S2 (and later also Sentinel-1/S1) images are available. For S2, twelve channels are available, and for S1, two channels are available. We only use three of the S2 channels (RGB images). Each image patch is annotated by multiple land-cover classes (i.e., multi-labels) taken from the CORINE Land Cover database of the year 2018 (CLC 2018). Originally, 43 labels were used. These were later merged into 19 labels [32].

### 2.2. Overview of the Learning Methods for Multi-Label Classification

We address the MLC task by applying a Convolutional Neural Network (CNN) as a deep learning approach and exploiting the the transfer learning paradigm. We use the

weights learned on the ImageNet as the initialization procedure, and we further perform calibration by fine-tuning the models on the training sets of of RSI. The deep models are used in two different scenarios, namely as end-to-end approaches to directly perform the MLC task (Figure 3a,b), and as CNN-based feature extractors (Figure 3a) to generate feature representations, which are used as inputs to the tree-ensemble methods such as Random Forests and Extremely Randomized Trees for MLC (Figure 3c).

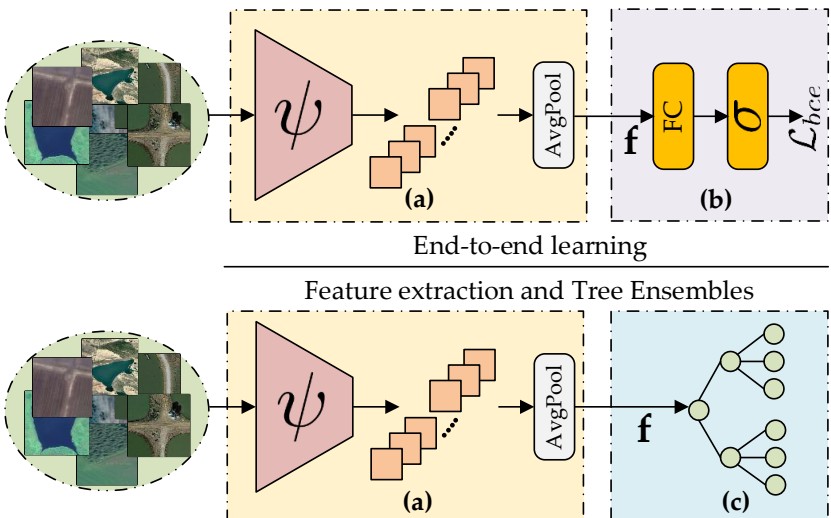

**Figure 3.** Overview of the proposed strategy for MLC of RSI. Part (**a**) of the strategy is the fully convolutional CNN backbone feature extractor. This part is same for both end-to-end learning, and feature extraction. The next part of the strategy, marked (**b**), is only used during end-to-end learning. The last part, marked with (**c**), is used for training tree ensemble methods from the extracted features, as a separate task and not a part of the end-to-end learning process. For this separate task the CNN feature extractor (**a**) is used in offline mode to generate the required feature representations, which serve as inputs to learn tree ensembles.

We make the following modifications to the used network architecture for the MLC task. To make it applicable to differently sized input images, we make sure the backbone CNN is fully convolutional. Next, to reduce the spatial dimension, the feature maps are aggregated through the average pooling layer, where $d$-dimensional feature representations **f** are extracted. The classification part is replaced with a single fully connected (FC) layer with the number of output units equal to the number of classes. This can also be seen as a layer providing non-normalized log probabilities $z_i$, i.e., scores (or also called *logit*s), that a classification model generates. This *logit* vector is then converted to a vector of probabilities by using the *sigmoid* activation function $\sigma(z_{i,c}) = 1/(1 + e^{-z_{i,c}})$, where $z_{i,c}$ is the *logit* of the predicted class. The *sigmoid* function is suggested for MLC, instead of the *softmax* function, as the activation of the last layer of the model, since the probabilities produced by *sigmoid* are independent and are not constrained to sum up to one. It follows a Bernoulli distribution and thus allows multiple label predictions [18]. It is important to mention that this part of the model (Figure 3b) is only used during training in the end-to-end mode to directly perform the MLC task.

The training process uses the standard binary cross-entropy learning objective, which takes the following form:

$$\mathcal{L}_{bce} = -\frac{1}{n}\sum_{i=1}^{n}[y_i log(\hat{y_i}) + (1 - y_i)log(1 - \hat{y_i})], \tag{1}$$

where $n$ is the number or samples in a given mini-batch, $y_i = [y_{i,1}, y_{i,2}, \ldots, y_{i,c}]$ is a binary vector representing the multi-labels of a given image, and $\hat{y_i}$ is the predicted output vector

of probabilities from the *sigmoid* layer. This learning objective is most commonly used in MLC tasks [18].

### 2.2.1. Deep Learning Methods

To investigate the impact of the network architecture on the learning process, we consider several popular CNN configurations pre-trained on ImageNet. We use VGG (VGG-16, VGG-19) [23] , ResNet (ResNet-34, ResNet-50, ResNet-152) [24] and EfficientNet (EfficientNet-B0, EfficientNet-B1, EfficientNet-B2) [25] as backbone CNN models for the MLC task. Throughout the experiments, these methods are used either in an end-to-end manner or as CNN-based feature extractors.

- VGGs: VGG is a deep CNN network architecture developed by the Visual Geometry Group (VGG) team [23]. It is also the basis of ground-breaking object recognition models that surpass baselines on many tasks and datasets beyond ImageNet and is still one of the most popular image recognition architectures. Two variants of this family of architectures are intensively studied for their performance—VGG-16 and VGG-19. The VGG-16 model can be seen as an upgrade of AlexNet, while VGG-19 is similar to VGG-16 but contains more layers. They are modeled in such a way that convolutions would actually look simpler by replacing large AlexNet convolution filters with a $3 \times 3$ filter, while padding to maintain the same size before a $2 \times 2$ *MaxPool* layer down-samples the image size.
- ResNets: ResNets are a family of deep CNN architectures that follow the residual learning principle to ease the training of very deep networks [24]. Their design offers an efficient way to solve the issues related to the vanishing gradients. ResNet follows VGG's full $3 \times 3$ convolutional layer design. The residual block has two $3 \times 3$ convolutional layers with the same number of output channels. Each convolutional layer is followed by a batch normalization layer and a Rectified Linear Unit (*ReLU*) activation function. Then, there is a skip (or so-called skip connection) of those two convolution operations, where the input is directly added before the final *ReLU* activation function. This kind of design requires that the output of the two convolutional layers has to be the same shape as the input, so that they can be added together. By configuring different numbers of channels and residual blocks in the module, we can create different ResNet models, such as the deeper 152-layer ResNets, i.e., ResNet-152. For the experiments, we use three variants of ResNet: ResNet-34, ResNet-50, and ResNet-152.
- EfficientNets: Unlike conventional deep CNNs, which are often over-parameterized, and arbitrarily scale network dimensions, such as width, depth, and resolution, EfficientNets are methods that uniformly scale each dimension with a fixed set of scaling coefficients [25]. These models surpass state-of-the-art accuracy, with up to 10 times better efficiency (i.e., are both smaller and faster than competitors).

### 2.2.2. Tree Ensemble Methods

In the experiments, we consider two types of ensemble methods based on decision trees for MLC as a main learning model, namely, ensembles based on random forests for MLC [33] and extremely randomized trees for MLC [34], respectively.

- Random Forest: Random forest (RF) is an ensemble learning method for classification and regression, which creates a set of individual decision trees that operate as an ensemble. It uses bagging and feature randomness to create diversity among the predictors: At each node in the decision tree, a random subset of attributes is taken, and the best split is selected from this subset of attributes. Each individual tree in the random forest provides a class prediction, where the predictions can be aggregated by taking the average (for regression tasks) and the majority or probability distribution vote (for classification tasks). RFs were adapted for the task of MLC [33].
- Extremely Randomized Trees: Extremely Randomized Trees, or so-called Extra Trees (ET), is also an ensemble learning method similar to the Random Forest, which is based on extreme randomization of the tree construction algorithm. As compared

to the Random Forest ensemble, it operates with two key differences: it splits nodes by choosing the cut-points fully at random, and it uses the whole learning sample to grow the trees. The randomness in this method comes from the random splits of all observations, rather then bootstrapping the data as in RF. ETs were adapted for the task of MLC [34].

## 3. Experimental Design

This section presents the details of the experimental study design. It includes a detailed overview of the experimental setup describing the different learning settings in the end-to-end approaches and the tree ensemble methods. Next, we describe the evaluation strategy of the partitioning of the image datasets into disjoint splits used for training and testing MLC models. Finally, we describe the evaluation measures used to assess the predictive performance of the different methods as well as the statistical procedures used to analyze the results.

The experimental study is tailored to answer the following research questions:

(i)     What is the influence of the learning strategy on the performance of end-to-end approaches: Is fine-tuning or pre-training only more suitable for solving the RSI-MLC task?

(ii)    Which network architecture is the best choice for end-to-end MLC of RSI and for use as a feature extractor and further training of tree ensembles for MLC?

(iii)   How do end-to-end learning and feature extraction plus tree ensembles compare on the task of RSI for MLC (assessed by using the best performing architecture from the previous analysis)? and

(iv)    How does the number of training examples influence the predictive performance of the methods used?

### 3.1. Experimental Setup

#### 3.1.1. End-to-End Learning Approaches

We train the deep network models by using two different learning strategies based on transfer learning. Both settings use the *ImageNet weights* for initialization, while fine-tuning different parts of the backbone CNN model. In the first learning setting, shown in Figure 4a, we fix all the model layers pre-trained on ImageNet and directly *t*ransfer the learned knowledge to the target domain. In the second learning setting, shown in Figure 4b, the entire network architecture is being fine-tuned to the new target domain. In both settings, we use a single application-dependent fully connected layer learned from scratch (i.e., the classification layer).

We use the binary cross entropy as an objective function (see Equation (1)) to optimize the model parameters, and apply the same training procedure and hyperparameters for 100 epochs across all datasets. The optimization of the model parameters is performed with the Adam optimizer with a learning rate of $1 \times 10^{-4}$ and a mini-batch of 64.

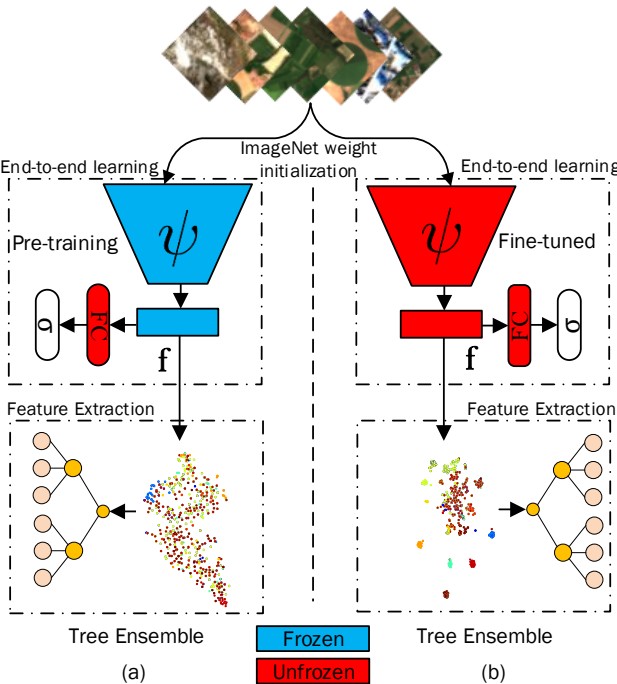

**Figure 4.** Two distinctive learning strategies for the MLC task. In the first learning setting, marked as (**a**), we rely on the ImageNet pre-training, where all the model layers are frozen, except the last fully connected layer, which is trained for the new target domain. In the second learning setting, marked as (**b**), the entire network architecture is fine-tuned along with the newly added fully connected layer. Moreover, we use both approaches either as feature extractors (in "offline" mode) jointly with tree ensemble methods, or as end-to-end learning machines to directly perform the MLC task. Note that the parts of the model highlighted with blue indicate that there is no update of the parameters in the model during the training process; on the other hand, with red, we highlight the opposite task, which means the parameters of the model are unfrozen and updated during training.

To prevent overfitting, we use data augmentation and modify data on the fly, so that our CNN models are transformation-invariant to the maximum possible extent. Moreover, we use data augmentation because we want to ensure that the predictive performance and generalization capability of the learned model is preserved to some extent, especially when training a predictive model on a dataset that contains a limited number of images (e.g., the Ankara dataset). Using the `Albumentations` library [35] (Available at: https://albumentations.ai (accessed on 4 January 2023 )), we performed the following image transformations: horizontal flipping, random rotations in the range ±10%, scaling by a factor in the range (0, 0.15), shifting by a factor in the range (0, 0.1), random crops with 50% of the original image size, random brightness within the range of (−0.3, 0.3), and random contrast within the range of (−0.3, 0.3). In addition, we consider the application of the following transformations to the input image: Contrast Limited Adaptive Histogram Equalization (CLAHE) to the input image, where the size of the grid for histogram equalization is set to $8 \times 8$ pixels; blurring with Gaussian kernel with $\sigma$ value in the range (3, 7); median blur with aperture linear size value $v$ randomly sampled once per image in the range (3, 7); motion blur with kernel size $\omega$ for blurring the input image, randomly chosen once per image in the range (3, 7); or Gaussian noise, where the variance range for noise is taken from the interval (10, 50).

The augmentations are performed in random order and with a 50% chance, which means that in such a setting there might be no augmentations applied over an image at all. This is important because, during the training process, the model needs to see the original image at least once in order to make reasonable predictions afterwards [36]. Some example augmentations are shown in Figure 5. Each row represents a specific dataset, where the left-most image marked with red is the original version, while the remaining images are the

augmented versions of the same image. Note that multiple augmentations could be applied on the same image. Because of that, we are not mixing certain image augmentations, such as the blurring operations (i.e., Gaussian, median and motion blur) with color (i.e., CLAHE) and Gaussian noise corruption transformations, simultaneously. The main reason for this is that we want to avoid strong image degradation and loss of contextual information, which is crucial in remote sensing imagery.

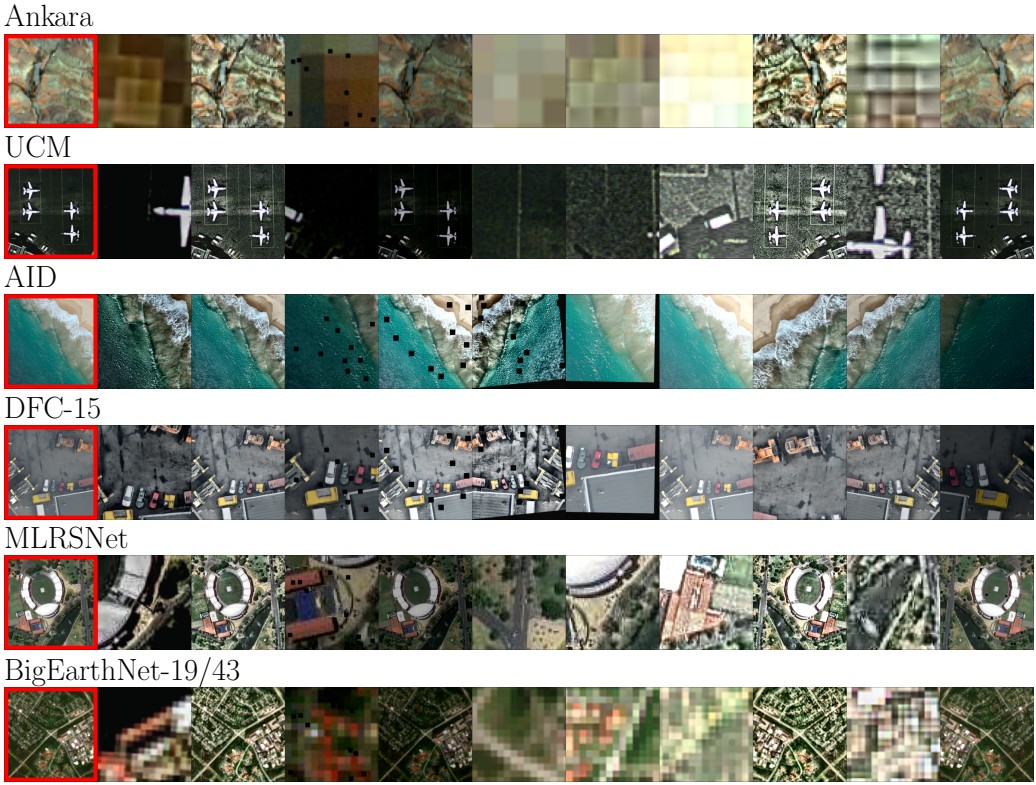

**Figure 5.** Augmentation examples. The left most image in each row marked with red is the original version, while the remaining images are the augmented versions. We can see the extent of variability added to the training datasets by the augmented images.

### 3.1.2. CNNs as Feature Extractors and Tree Ensembles

The deep learning methods presented in Section 2.2.1, are further used as feature extractors, for each of the two learning settings described in Section 3.1.1. Hence, we rely on two feature representations $\mathbf{f} \in \mathbb{R}^d$ extracted from the CNN feature extractors: representations based on pre-training (Figure 4a), and fine-tuning (Figure 4b). Furthermore, each feature representation is used as input to the tree ensemble methods, i.e., Random Forests (RF) and Extremely Randomized Trees (ERT)/Extra Trees (ET) for short. We use 150 base models in each of the ensembles and sqrt as the feature subset size.

### 3.2. Evaluation Strategy

We conducted experiments over the several publicly available RSI datasets described above and explored how the performance of the learning strategies is affected by data availability, image quality, and label dimensionality. To tackle this problem, we included datasets which are less challenging in terms of image resolution, such as DFC-15 and MLRSNet, and datasets such as Ankara, which is very limited in size and has poor image quality. To evaluate the effectiveness of the methods, we perform the splitting according to the datasets that already contain a predefined subset of images, such as AID and DFC-15, where the train and test ratios are approximately 80% and 20%. The remaining dataset is partitioned accordingly, by adopting an iterative stratified sampling strategy in order to preserve the relative frequency of the labels in the datasets to the maximum possible extent.

Moreover, we split a validation set of 10% from the overall training data, which is only used to monitor the learning progress and to provide an unbiased estimate of the model fit when tuning hyperparameters. A description of the datasets after the splitting procedure is given in Table 1.

### 3.3. Evaluation Measures and Statistical Analysis

Many evaluation measures are used to assess the predictive performance and effectiveness of MLC methods, offering different viewpoints on the performance of the methods. The evaluation measures can be grouped into two groups: measures based on bipartitions (example-based and label-based measures) and ranking-based evaluation measures [8]. The example-based evaluation measures compute the average difference between the true labels and the predicted labels for each data point, averaged across all the examples in the dataset. Unlike example-based measures, label-based measures evaluate each label separately and then average the performances across all labels. The ranking-based evaluation measures compare the predicted ranking of the labels with the ground truth ranking (where all present labels are ranked before all absent labels).

In this study, we present the results in terms of ranking-loss as an evaluation measure. Ranking loss evaluates the average fraction of label pairs that are misordered for a given example. Note, however, that we provide complete results in terms of all other evaluation measures for the MLC task in the Appendix A section (for the calculation of the evaluation measures, we use the `scikit-learn` implementation [37]). We focus on ranking loss, since we believe it is one of the best indicators for measuring the performance of methods for MLC. Moreover, this measure is threshold-independent, which means we do not rely on the use of techniques for threshold estimation to produce the predicted labels. Ranking loss is defined as follows:

$$rl = \frac{1}{N} \sum_{i=1}^{N} \sum_{(j,k):y_j > y_k} \left( I[r_i(j) < r_i(k)] + \frac{1}{2} I[r_i(j) = r_i(k)] \right), \tag{2}$$

where $y_i$ and $\hat{y}_i$ are the true and the predicted labels, respectively, $N$ is the number of examples, $r_i(j)$ is the ranking of label $j$ for instance $x_i$, and $I$ is an indicator function. The smaller the value of $rl$, the better the performance.

We use the Friedman test to assess whether the overall differences in performance of the used approaches evaluated across the RSI datasets are statistically significant and the post hoc Nemenyi test to detect between which methods the statistically significant differences occur. The obtained results are presented in the form of Nemenyi post hoc average rank diagrams [38] for the ranking loss measure. In the analysis, the significance level was set to $\alpha = 0.05$. The best-performing methods are on the left-most side of the diagram along the axis (average ranks closer to 1), and the methods whose predictive performance does not differ significantly at $\alpha = 0.05$, are connected with a red line.

### 3.4. Implementation Details

We implemented the deep learning models in the `PyTorch` framework (Available at: https://pytorch.org, accessed on 4 January 2023 ). We use the built-in implementations of VGGs, ResNets and EfficientNets, initialized with weights learned on ImageNet. We modify each of the backbone models in order to accept arbitrary-sized input images by replacing the last max pooling layer with the average pooling operation with a kernel size of 1, resulting in the following $d$-dimensional feature representations: $d = 4096$ for VGG-16 and VGG-19, $d = 512$ for ResNet-34, and $d = 2048$ for ResNet-50 and ResNet-152. We have 1280 output dimensions for EfficientNet-B0 and EfficientNet-B1, while EfficientNet-B2 has 1408 dimensions. Finally, for the tree ensemble methods, we use the `scikit-learn` [37] implementation of Random Forest tree ensembles for multi-label classification (MLC) [33] and Extra Tree ensembles for MLC [34]. The tree ensembles for MLC simultaneously predict the probability of each of the multiple class labels. The labels can then be ranked based on

these predicted probabilities. The complete source code and the datasets used to execute the study are publicly and freely available at https://github.com/marjanstoimchev/RSMLC (accessed on 4 January 2023).

## 4. Results and Discussion

This section presents the results of our experimental study in MLC of RSI. It answers one of the experimental questions posed in Section 3 in each subsection. It thus discusses (1) the influence of the learning strategy, (2) the comparison of different network architectures used as end-to-end approaches, as well as feature extractors, (3) the comparison between end-to-end methods and tree ensembles, and (4) the influence of the number of available labeled images on predictive performance.

### 4.1. The Influence of the Learning Strategy

In the first experiment, we explored whether fine-tuning or pre-training only is the more appropriate learning strategy (as defined in Section 3.1.1). To provide the answer to this question, we conducted experiments by training different network architectures over the datasets, and using them in two different modes of operation: (1) as feature extractors providing the feature representations to the tree ensembles, and (2) as end-to-end learning methods. Moreover, for each learning approach, we present the difference in performance in the form of a heat map, which can be formally defined as follows:

$$\mathbf{H} = \begin{bmatrix} rl_{11} & rl_{12} & \cdots & rl_{1n} \\ rl_{21} & rl_{22} & \cdots & rl_{2n} \\ \vdots & \vdots & \ddots & \vdots \\ rl_{m1} & rl_{m2} & \cdots & rl_{mn} \end{bmatrix} \in \mathbb{R}^{m \times n}, \tag{3}$$

where $rl_{i,j} = rl_{i,j}^{fine-tune} - rl_{i,j}^{pre-train}$ is the difference between fine-tuning ($rl^{fine-tune}$) and pre-training ($rl^{pre-train}$) in terms of the ranking loss measure, calculated for the $i$-th dataset and the $j$-th method, respectively. This is performed for $i = 1, \ldots, m$ and $j = 1, \ldots, n$, where $m$ denotes the number of datasets and $n$ is the number of methods/CNN architectures.

These differences in performance are presented in Figure 6. We observe that for smaller datasets (e.g., Ankara, UCM, etc.), pre-training is the preferred choice, which can be mostly seen for the VGG architectures trained in an end-to-end manner over the DFC-15 dataset, where the largest differences in ranking loss are observed. These differences are slightly smaller for the DFC-15 dataset and the tree-ensemble methods. On the other hand, the fine-tuned versions of the models perform significantly better when data availability is not a problem. Furthermore, for the tree ensemble methods, the differences in performance are significantly increased (and more on the positive side), which points to the fact that the quality of the feature representations is of great importance for the tree ensemble methods to further boost their predictive performance. Overall, we can conclude that by solely relying on the ImageNet pre-training, we end up with worse model performance in almost all cases, because the content present in RSI is quite complex as compared to images present in the ImageNet dataset. More detailed results of the analysis are presented in Table A1 (Appendix A), where the performance figures for the ranking loss measure are given.

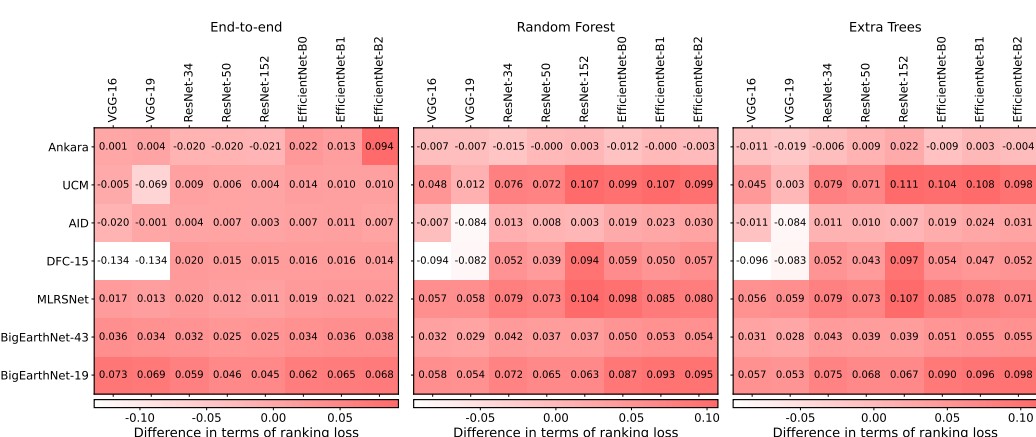

**Figure 6.** Comparison between fine-tuning and pre-training in terms of ranking loss. The results for the different architectures are presented in the form of a heat map showing the difference in performance between the learning approaches. Negative values indicate that pre-training is better than fine-tuning and positive when fine-tuning is better than pre-training. The intensities of the colors in the heat map are directly related to the difference in performance between the learning approaches (CNN architectures). The datasets on the y-axis are ordered by the number of images they contain.

### 4.2. Comparison of Different Network Architectures

Based on the analysis from Section 4.1, we used the fine-tuning approach to further explore which network architecture is the most suitable choice for RSI MLC tasks. The results are shown in terms of ranking loss and in the form of average rank diagrams. From Figure 7, we see that the EfficientNet variants, especially EfficientNet-B2, tend to produce the better results as compared to other network architectures. The differences in performance are statistically significant as compared to the VGG variants. The results also reveal that the EfficientNet-B2 model, used as base feature extractor in combination with tree ensemble methods, such as RF as in Figure 7b and ET in Figure 7c, is the clear winner, which indicates that this network architecture is the best choice for this task. Moreover, the ResNet-based network architectures are the closest competitors to EfficientNet variants (both when used with tree ensembles and in an end-to-end manner), i.e., the ResNet-152 model. Although the EfficientNet variants are not statistically significantly better than ResNet-152, they are by far more lightweight in terms of model parameters (e.g., EfficientNet-B0 is approximately 11× smaller, EfficientNet-B1 is 8×, and EfficientNet-B2 is approximately 6.5× smaller than ResNet-152, respectively). Comparison between different network architectures in terms of other MLC performance measures in the form of average rank diagrams is given in Figures A1–A6 in the Appendix A (Figures A1–A3 for fine-tuned features and label-based, example-based and ranking-based measures, respectively, Figures A4–A6 for pre-trained features and label-based, example-based and ranking-based measures, respectively).

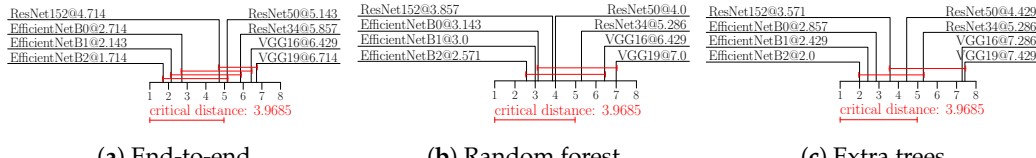

| (**a**) End-to-end | (**b**) Random forest | (**c**) Extra trees |

**Figure 7.** Comparison between different network architectures in terms of ranking loss. The results are presented in the form of average rank diagrams at 0.05 significance level for (**a**) End-to-end learning, (**b**) Random forests and (**c**) Extra trees. The best ranking methods are at the left-most side of the diagram. The difference in performance among the methods connected with a red line is not statistically significant.

Overall, we can conclude that in the fine-tuning setting, it is favorable to use lightweight network architectures in terms of model parameters for end-to-end learning, which are

also capable of learning more discriminative feature representations when used as feature extractors. This is in direct relation to their capability of capturing high-level content present in RSI to a greater extent as compared to the other network architectures. Moreover, they are the preferred choice when addressing the problems encountered in challenging deployment scenarios, which means they can maintain good predictive performance while keeping the computational costs at a reasonable level.

### 4.3. Comparison of Different Learning Approaches

To answer the experimental question of how end-to-end learning and feature extraction plus the tree ensembles compare in the task of RSI MLC, we present the results in the form of average rank diagrams, where we use the pre-trained and fine-tuned versions of the EfficientNet-B2 model. Recall that EfficientNet-B2 produced the best results overall, either when used as a feature extractor where the extracted feature representations are further utilized in the tree ensembles, or as an end-to-end approach to directly address the MLC task. The results are shown in Figure 8a for fine-tuning and in Figure 8b for pre-training only.

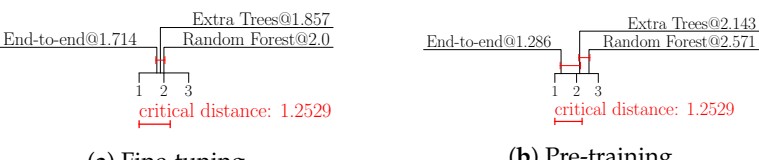

(**a**) Fine-tuning  (**b**) Pre-training

**Figure 8.** Comparison between different learning methods in terms of ranking loss. The results are presented in the form of average rank diagrams at 0.05 significance level for (**a**) Fine-tuning and (**b**) Pre-training only. The best ranking methods are at the left-most side of the diagram. The differences among the methods connected with a red line is not statistically significant.

As seen from the diagrams, end-to-end learning is ranked best, followed by Extra trees and Random Forests. In the case of fine-tuning (Figure 8a), there is no statistically significant difference among the classification methods. In the pre-training setting, the end-to-end learning approach is significantly better than the random forest method. Overall, the differences in predictive performance depend on the specific deep neural network architecture and the learning setting. Comparison between different learning methods in terms of other MLC performance measures in the form of average rank diagrams is given in Figures A7 and A8 in the Appendix A. Figure A7 concerns the use of fine-tuned and Figure A8 the use of pre-trained features.

### 4.4. Influence of the Number of Available Labeled Images

To answer the fourth question about the relation of the number of available images and the performance of the different MLC models, we conducted additional experiments focusing on the biggest dataset available—BigEarthNet. We design an experimental protocol which partitions the BigEarthNet dataset into distinctive subsets of images, namely, fractions of: 0.1%, 0.5%, 1%, 5%, 10%, 25%, and 50% of 590,326 images, respectively. Each of the fractions is further split into disjoint subsets of training, validation, and testing images, with sizes 70%, 10%, and 20% of the fraction size, respectively. Moreover, the test sets are built in a cumulative manner, which means the test subset of images from the previous fractions are inherited into the test set of the next fraction. By doing this, we are evaluating the effectiveness of the models on new subsets of images, while taking the old ones into account, i.e., we simulate a scenario where we add new images. The sampling of the fractions and the subsets within the fractions can be done in two different ways: (i) sampling with stratification, and (ii) random sampling. Furthermore, we are also assessing the generalization capability of the model when exposed to different distribution shifts. To obtain a more reliable estimate of the predictive performance, we repeated the experiment five times and calculated the average performance and standard deviation $(\mu \pm \sigma)$ in terms of ranking loss.

To carry out the experiment, we used the two versions of the BigEarthNet dataset, namely, BigEarthNet with 19 and 43 CLC nomenclatures. We selected this dataset because it is the largest one in terms of the number of images. For these experiments, we used EfficientNet-B2 as network architecture, since it produced the best results overall in the previous experiments. We fine-tuned the model parameters for 25 epochs and used the trained model as a feature extractor for the tree ensembles, as well as an end-to-end approach. We present the results in the form of learning curves in order to see how the number of training examples influences the predictive performance. The results of the experiment for the BigEarthNet-19 and BigEarthNet-43 are shown in Figure 9. We can see that the performance of the learning methods in all cases improves of the total number of training examples by up to 10%, after which it degrades. End-to-end learning methods perform worse than tree ensemble methods in all cases in the learning curve, but the differences in performance are only visible in the case of stratified sampling. The differences are more expressed in the BigEarthNet-19 dataset (Figure 9). This means that the different learning methods are affected differently by the choice of the sampling strategy.

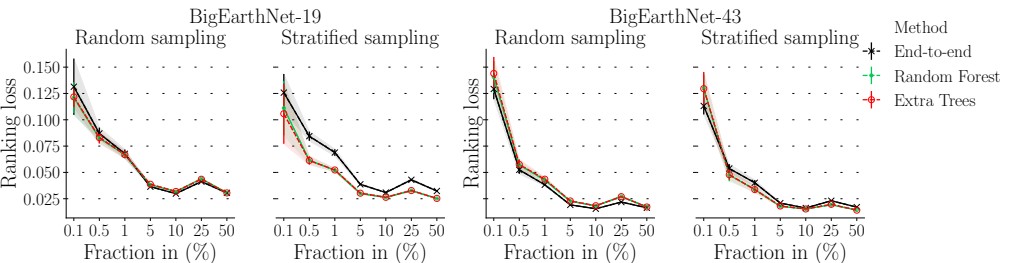

**Figure 9.** Comparison between different learning methods in terms of ranking loss when evaluated on different fractions of labeled examples from the BigEarthNet-19 and BigEarthNet-43 datasets,. The results are shown as learning curves depicting $\mu \pm \sigma$ for ranking loss across five repeats. The y-axis is shared across the figures.

## 5. Conclusions

In this study, we have presented a comparative analysis of methods for multi-label classification of remote sensing imagery. We have compared several popular deep learning methods based on two modes of operation, where they are (1) used as feature extractors in a combination with tree ensemble methods such as random forests and extra trees, and (2) used as end-to-end approaches to directly address the MLC task.

We focused on four aspects: (1) different transfer learning paradigms, namely, learning features based on ImageNet pre-training only, as well as learning features with fine-tuning, where the whole network architecture and the model parameters are trained on the new target domain of interest; (2) comparison between different network architectures; (3) comparison between tree ensembles and end-to-end approaches; and (4) investigating the influence of the number of labeled examples on the relative predictive performance of MLC methods. In the first dimension, we showed that it is beneficial to fine-tune the models on RSI to improve their performance. However, in certain cases, where the number of data is limited, it is better to extract features with ImageNet pre-training and only fine-tune the last layer for classification. Furthermore, we showed that it is very important to choose a proper network architecture: EfficientNets proved to be overall the most suitable choice for the task of MLC. They have significantly fewer parameters as compared to the ResNet variants, and no statistically significant difference in predictive performance is observed. We also showed that having the right feature extractor plays an important role in boosting the performance of tree ensemble methods, so that they outperform the end-to-end approaches in certain cases. In the last dimension, we investigated the influence of the amount of labeled data from the BigEarthNet dataset on the relative performance of MLC methods, where we applied two types of sampling strategies: random sampling and sampling with stratification. We showed that in such a setting, the tree ensemble methods

outperform the end-to-end approaches, with the difference in performance clearly visible for stratified sampling.

Considering the findings from this study, further extension of this work should focus on several aspects. Firstly, we will incorporate even a wider range of deep learning models specially devised for the RS MLC task. Next, we will take into account different MLC loss functions to analyze whether they are more suitable for the RS MLC task. Moreover, since the label space of BigEarthNet dataset is organized in a hierarchical manner, we will further analyze the effect of using the hierarchical information in the tree ensemble methods and in the end-to-end approaches. Lastly, we will focus on the application of the methods in a semi-supervised learning setting, where we will exploit the abundance of the unlabeled data in RS domain and investigate if the semi-supervised counterparts can surpass the performance of the supervised learning methods.

**Author Contributions:** Conceptualization, S.D.; methodology, M.S., D.K., and S.D.; software, M.S.; validation, M.S. and D.K.; writing—original draft preparation, M.S.; writing—review and editing, D.K. and S.D.; supervision, S.D. All authors have read and agreed to the published version of the manuscript.

**Funding:** We acknowledge the support of the European Space Agency ESA through the activity of the AiTLAS - AI4EO rapid prototyping environment. This work was also partially supported by the Slovenian research agency through the knowledge technologies program P2-0103 and the project J2-2505.

**Institutional Review Board Statement:** Not applicable.

**Informed Consent Statement:** Not applicable.

**Data Availability Statement:** The AID Dataset in this study is openly and freely available at https://github.com/Hua-YS/AID-Multilabel-Dataset (accessed on 1 July 2022). The DFC-15 dataset in this study is openly and freely available at https://drive.google.com/drive/folders/1TKGS6 TIRxQ6a7gdaj0cHs-mRCtv_J1HA (accessed on 1 July 2022). The MLRSNet dataset in this study is openly and freely available at https://data.mendeley.com/datasets/7j9bv9vwsx/2 (accessed on 1 July 2022). UCM dataset in this study is openly and freely available at https://drive.google.com/file/d/1DtKiauowCB0ykjFe8v0OVvT76rEfOk0v/view (accessed on 1 July 2022). The Ankara dataset in this study is openly and freely available at https://bigearth.eu/datasets (accessed on 1 July 2022). The BigEarthNet-19 and BigEarthNet-43 in this study are openly and freely available at https://bigearth.net/ (accessed on 1 July 2022).

**Conflicts of Interest:** The authors declare no conflict of interest.

## Abbreviations

The following abbreviations are used in this manuscript:

| | |
|---|---|
| RSI | Remote Sensing Images |
| MLC | Multi-Label-Classification |
| DNN | Deep Neural Network |
| CNN | Convolutional Neural Network |
| DL | Deep Learning |
| RNN | Recurrent Neural Network |
| LSTM | Long Short-Term Memory |
| GCN | Graph Convolutional Network |
| CLC | Corine Land Cover |
| ReLU | Rectified Linear Unit |
| RF | Random Forest |
| ET | Extra Trees |

## Appendix A. Complete Results from the Experimental Evaluation

**Table A1.** The performance in terms of ranking loss measure of fine-tuning and pre-training feature learning approaches with different network architectures and different MLC approaches. The best performing network architecture for each dataset is highlighted with green.

| Approach | Datasets | VGG-16 | VGG-19 | ResNet-34 | ResNet-50 | ResNet-152 | EfficientNet-B0 | EfficientNet-B1 | EfficientNet-B2 |
|---|---|---|---|---|---|---|---|---|---|
| | | | | | | Pre-Training | | | |
| | Ankara | 0.298 | 0.371 | 0.343 | 0.351 | 0.350 | 0.349 | 0.330 | 0.422 |
| | UCM | 0.186 | 0.180 | 0.154 | 0.149 | 0.135 | 0.194 | 0.185 | 0.184 |
| | AID | 0.215 | 0.208 | 0.171 | 0.181 | 0.179 | 0.198 | 0.194 | 0.188 |
| End-to-end | DFC-15 | 0.176 | 0.176 | 0.147 | 0.134 | 0.134 | 0.127 | 0.120 | 0.113 |
| | MLRSNet | 0.347 | 0.360 | 0.306 | 0.240 | 0.229 | 0.318 | 0.300 | 0.326 |
| | BigEarthNet-19 | 0.557 | 0.550 | 0.461 | 0.399 | 0.391 | 0.460 | 0.476 | 0.478 |
| | BigEarthNet-43 | 0.578 | 0.546 | 0.480 | 0.431 | 0.410 | 0.468 | 0.481 | 0.480 |
| | Ankara | 0.322 | 0.320 | 0.324 | 0.329 | 0.388 | 0.317 | 0.337 | 0.345 |
| | UCM | 0.381 | 0.398 | 0.469 | 0.420 | 0.539 | 0.495 | 0.508 | 0.468 |
| | AID | 0.250 | 0.244 | 0.265 | 0.247 | 0.294 | 0.268 | 0.257 | 0.262 |
| Random Forest | DFC-15 | 0.297 | 0.337 | 0.235 | 0.201 | 0.342 | 0.259 | 0.222 | 0.242 |
| | MLRSNet | 0.529 | 0.545 | 0.566 | 0.549 | 0.615 | 0.587 | 0.567 | 0.548 |
| | BigEarthNet-19 | 0.534 | 0.525 | 0.588 | 0.543 | 0.532 | 0.637 | 0.670 | 0.669 |
| | BigEarthNet-43 | 0.547 | 0.537 | 0.600 | 0.557 | 0.545 | 0.654 | 0.686 | 0.683 |
| | Ankara | 0.318 | 0.312 | 0.331 | 0.325 | 0.370 | 0.328 | 0.320 | 0.330 |
| | UCM | 0.390 | 0.405 | 0.457 | 0.417 | 0.552 | 0.483 | 0.513 | 0.462 |
| | AID | 0.254 | 0.250 | 0.255 | 0.253 | 0.305 | 0.265 | 0.250 | 0.256 |
| Extra Trees | DFC-15 | 0.297 | 0.338 | 0.235 | 0.204 | 0.351 | 0.235 | 0.218 | 0.229 |
| | MLRSNet | 0.530 | 0.545 | 0.567 | 0.549 | 0.620 | 0.573 | 0.556 | 0.534 |
| | BigEarthNet-19 | 0.539 | 0.528 | 0.608 | 0.567 | 0.556 | 0.658 | 0.693 | 0.698 |
| | BigEarthNet-43 | 0.550 | 0.540 | 0.622 | 0.581 | 0.570 | 0.676 | 0.709 | 0.713 |
| | | | | | | Fine-tuning | | | |
| | Ankara | 0.294 | **0.285** | 0.377 | 0.356 | 0.360 | 0.335 | 0.353 | 0.322 |
| | UCM | 0.224 | 0.508 | 0.101 | 0.097 | 0.112 | 0.088 | 0.096 | **0.081** |
| | AID | 0.265 | 0.202 | 0.152 | 0.143 | 0.147 | 0.137 | **0.131** | 0.137 |
| End-to-end | DFC-15 | 0.433 | 0.433 | 0.068 | 0.075 | 0.067 | 0.054 | 0.046 | 0.050 |
| | MLRSNet | 0.180 | 0.223 | 0.093 | 0.091 | 0.088 | **0.082** | 0.084 | 0.084 |
| | BigEarthNet-19 | 0.276 | 0.282 | 0.235 | 0.236 | 0.210 | 0.207 | 0.203 | **0.202** |
| | BigEarthNet-43 | 0.271 | 0.276 | 0.243 | 0.232 | 0.199 | 0.206 | 0.195 | **0.194** |
| | Ankara | 0.318 | 0.319 | 0.344 | 0.338 | 0.345 | 0.304 | 0.335 | 0.320 |
| | UCM | 0.182 | 0.323 | 0.103 | 0.098 | 0.103 | 0.106 | 0.104 | 0.106 |
| | AID | 0.245 | 0.197 | 0.146 | 0.144 | 0.149 | 0.138 | 0.138 | 0.137 |
| Random Forest | DFC-15 | 0.433 | 0.433 | 0.050 | 0.047 | 0.050 | 0.046 | **0.041** | 0.044 |
| | MLRSNet | 0.185 | 0.221 | 0.104 | 0.103 | 0.102 | 0.093 | 0.095 | 0.090 |
| | BigEarthNet-19 | 0.258 | 0.268 | 0.229 | 0.219 | 0.214 | 0.222 | 0.219 | 0.217 |
| | BigEarthNet-43 | 0.255 | 0.267 | 0.235 | 0.230 | 0.219 | 0.228 | 0.221 | 0.224 |
| | Ankara | 0.364 | 0.348 | 0.342 | 0.322 | 0.362 | 0.313 | 0.343 | 0.321 |
| | UCM | 0.177 | 0.334 | 0.103 | 0.102 | 0.102 | 0.100 | 0.098 | 0.097 |
| | AID | 0.248 | 0.194 | 0.144 | 0.146 | 0.146 | 0.135 | 0.134 | 0.136 |
| Extra Trees | DFC-15 | 0.433 | 0.433 | 0.049 | 0.050 | 0.046 | 0.046 | 0.041 | 0.043 |
| | MLRSNet | 0.184 | 0.221 | 0.106 | 0.104 | 0.103 | 0.097 | 0.100 | 0.094 |
| | BigEarthNet-19 | 0.257 | 0.268 | 0.227 | 0.217 | 0.213 | 0.222 | 0.218 | 0.216 |
| | BigEarthNet-43 | 0.255 | 0.267 | 0.233 | 0.228 | 0.217 | 0.227 | 0.219 | 0.224 |

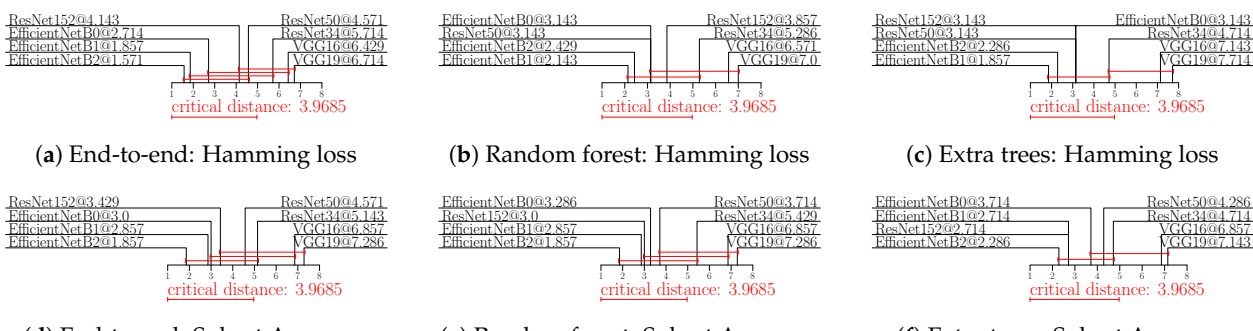

**(a)** End-to-end: Micro-F1

**(b)** Random forest: Micro-F1

**(c)** Extra trees: Micro-F1

**(d)** End-to-end: Micro-recall

**(e)** Random forest: Micro-recall

**(f)** Extra trees: Micro-recall

**(g)** End-to-end: Micro-precision

**(h)** Random forest: Micro-precision

**(i)** Extra trees: Micro-precision

**(j)** End-to-end: Macro-F1

**(k)** Random forest: Macro-F1

**(l)** Extra trees: Macro-F1

**(m)** End-to-end: Macro-recall

**(n)** Random forest: Macro-recall

**(o)** Extra trees: Macro-recall

**(p)** End-to-end: Macro-precision

**(q)** Random forest: Macro-precision

**(r)** Extra trees: Macro-precision

**Figure A1.** Performance of different network architectures in terms of label-based evaluation measures for fine-tuned features. The results are presented in the form of average rank diagrams at a 0.05 significance level. The best ranking methods are at the left-most side of the diagram. The difference among the methods connected with a red line is not statistically significant.

**(a)** End-to-end: Hamming loss

**(b)** Random forest: Hamming loss

**(c)** Extra trees: Hamming loss

**(d)** End-to-end: Subset Accuracy

**(e)** Random forest: Subset Accuracy

**(f)** Extra trees: Subset Accuracy

**Figure A2.** Performance of different network architectures in terms of example-based evaluation measures for fine-tuned features. The results are presented in the form of average rank diagrams at a 0.05 significance level. The best ranking methods are at the left-most side of the diagram. The difference among the methods connected with a red line is not statistically significant.

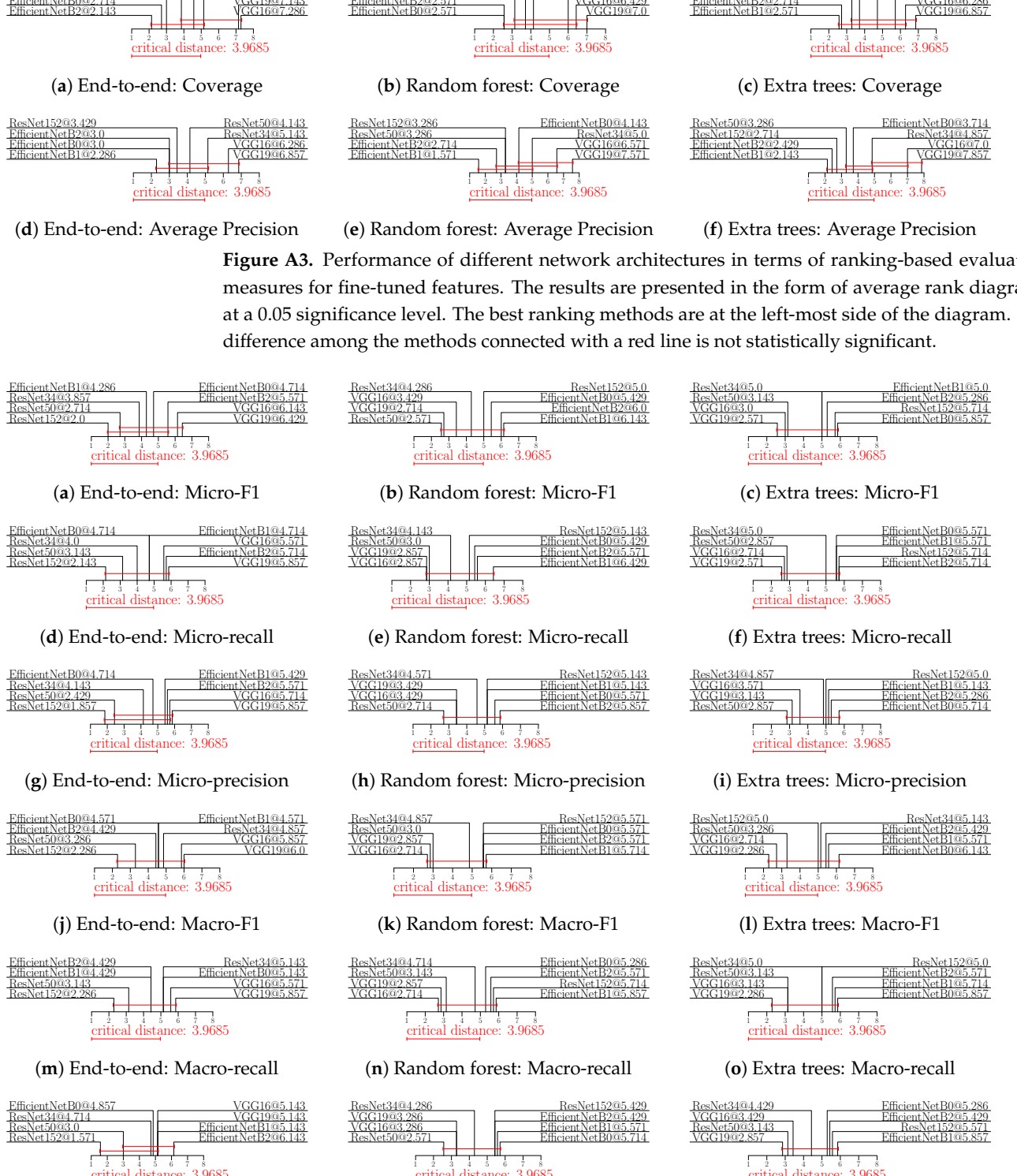

**Figure A3.** Performance of different network architectures in terms of ranking-based evaluation measures for fine-tuned features. The results are presented in the form of average rank diagrams at a 0.05 significance level. The best ranking methods are at the left-most side of the diagram. The difference among the methods connected with a red line is not statistically significant.

**Figure A4.** Performance of different network architectures in terms of label-based evaluation measures for pre-trained features. The results are presented in the form of average rank diagrams at a 0.05 significance level. The best ranking methods are at the left-most side of the diagram. The difference among the methods connected with a red line is not statistically significant.

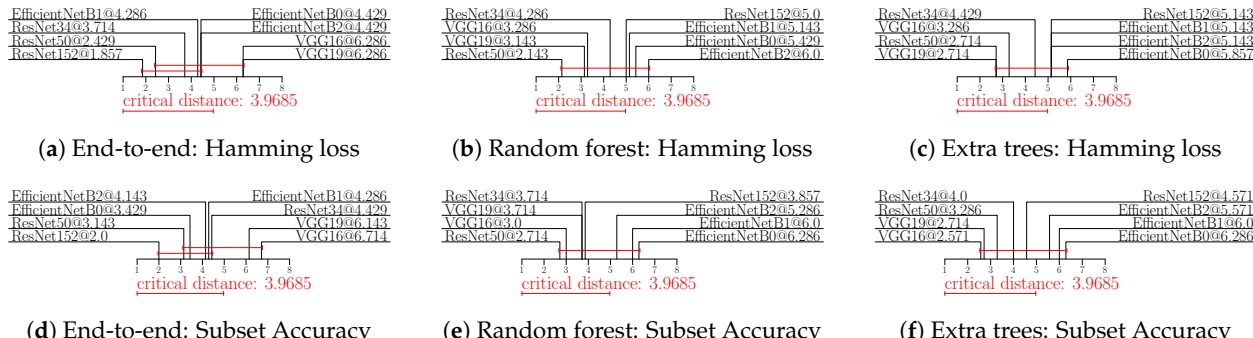

**Figure A5.** Performance of different network architectures in terms of example-based evaluation measures for pre-trained features. The results are presented in the form of average rank diagrams at a 0.05 significance level. The best ranking methods are at the left-most side of the diagram. The difference among the methods connected with a red line is not statistically significant.

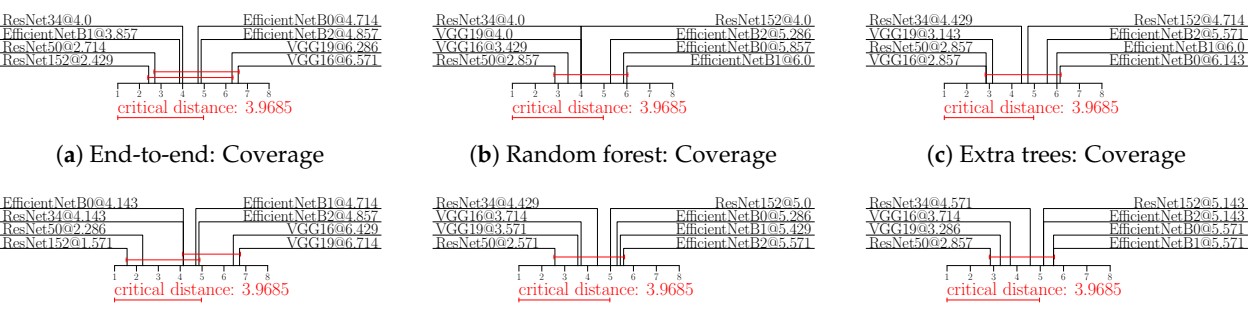

**Figure A6.** Performance of different network architectures in terms of ranking-based evaluation measures for pre-trained features. The results are presented in the form of average rank diagrams at a 0.05 significance level. The best ranking methods are at the left-most side of the diagram. The difference among the methods connected with a red line is not statistically significant.

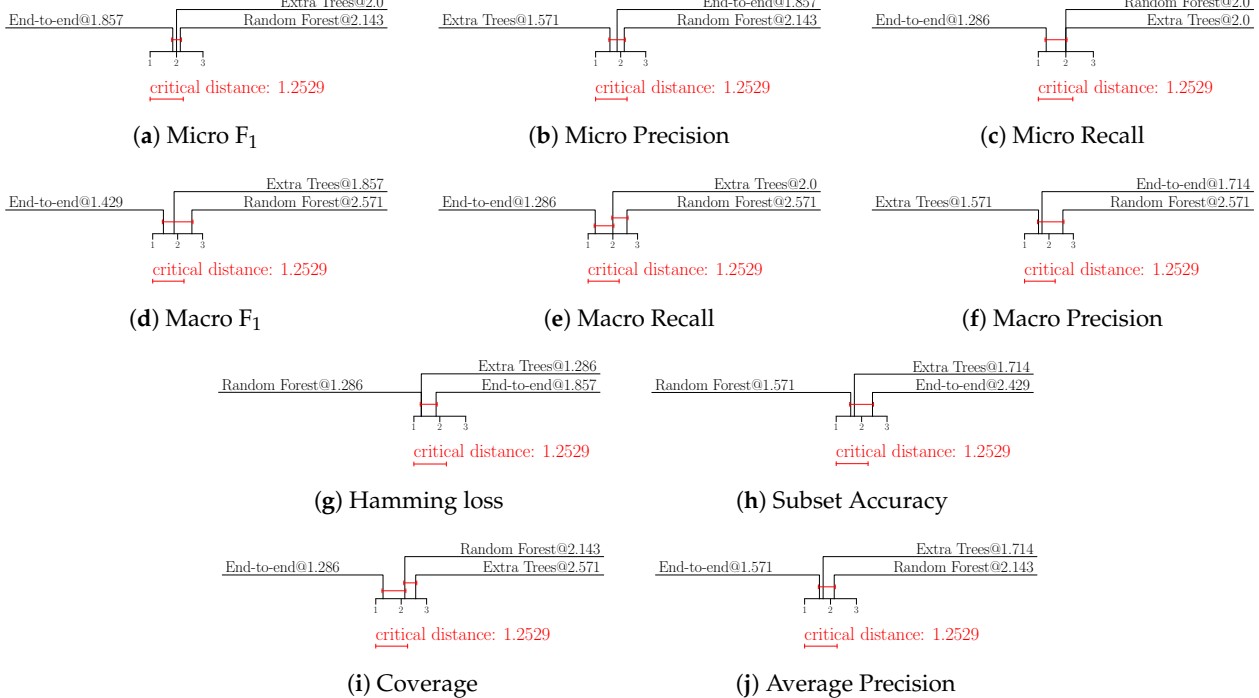

**Figure A7.** Performance of different learning methods in terms of all MLC evaluation measures for fine-tuned features. The results are presented in the form of average rank diagrams at a 0.05 significance level. The best ranking methods are at the left-most side of the diagram. The difference among the methods connected with a red line is not statistically significant.

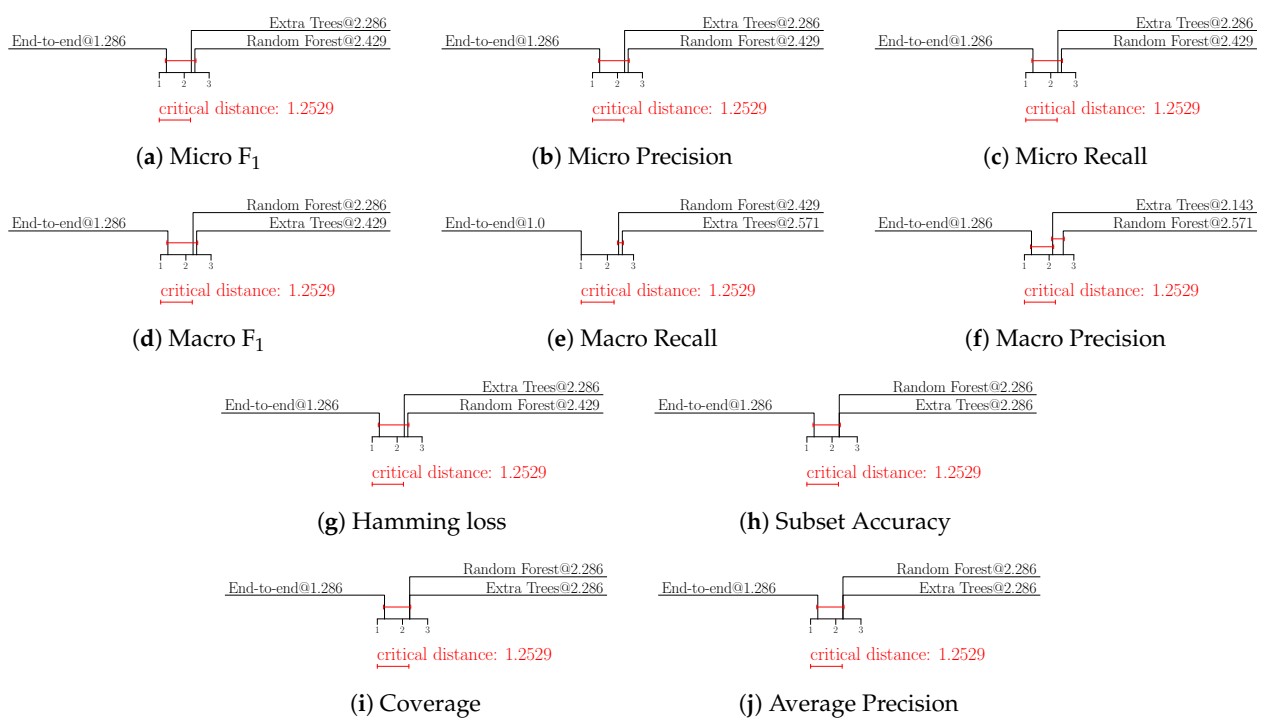

**Figure A8.** Performance of different learning methods in terms of all MLC evaluation measures for pre-trained features. The results are presented in the form of average rank diagrams at a 0.05 significance level. The best ranking methods are at the left-most side of the diagram. The difference among the methods connected with a red line is not statistically significant.

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
