# Peer review of "Deep Network Architectures as Feature Extractors for Multi-Label Classification of Remote Sensing Images"

_remotesensing, doi:10.3390/rs15020538_

Round 1
Reviewer 1 Report
Thank you for your efforts.
In general, I believe you are presenting a fact that would be obvious before. You are comparing feature extractors against fine-tuning and concluding that tune-tuning is better when we have enough label data. What is the theoretical questions that you wanted to address in your experiments? Have had any other expectations before your experiments? There are plenty of papers in the deep learning field discussing how transfer learning can be used to fine-tune a network to a new application domain from a theoretical perspective. When you have a totally different application than pre-trained networks (imageNet), obviously, extracted features are not relevant to the new application and you need to fine-tune all or part of the feature extraction part of the network. Some papers especially discuss these issues for a specific application: Sea ice classification of SAR imagery based on convolution neural networks
or
Convolutional Neural Network for Remote-Sensing Scene Classification: Transfer Learning Analysis
Deep Transfer Learning for Crop Yield Prediction with Remote Sensing Data
In the conclusion section, considering limited label data, You say "it is better to use the ImageNet pre-training and only fine-tune the last layer for classification" (521). Do you have any experiments to show this? Have you done any experiments to show which and how many layers should be fine-tuned to get better results?
Random augmentation on none RGB images, especially when different channels are statistically dependent is arguable. You are discussing a lot about augmentation and also put a figure to show the augmentation outputs! Could you please clarify how did you deal with this issue when you are using Hyperspectral datasets?
Your main message in this paper is about the different strategies to fine-tune a network for a new application (RS). It is expected to review relevant papers in the introduction section. I think you can improve the introductions section. Some papers mentioned such as attention mechanism, LSTM, ... not sure if relevant. Looks random.
Best Regards
Good Luck
Author Response
REVIEWER 1
Thank you for your efforts.
In general, I believe you are presenting a fact that would be obvious before. You are comparing feature extractors against fine-tuning and concluding that tune-tuning is better when we have enough label data. What is the theoretical questions that you wanted to address in your experiments? Have had any other expectations before your experiments? There are plenty of papers in the deep learning field discussing how transfer learning can be used to fine-tune a network to a new application domain from a theoretical perspective. When you have a totally different application than pre-trained networks (imageNet), obviously, extracted features are not relevant to the new application and you need to fine-tune all or part of the feature extraction part of the network. Some papers especially discuss these issues for a specific application: Sea ice classification of SAR imagery based on convolution neural networks or Convolutional Neural Network for Remote-Sensing Scene Classification: Transfer Learning Analysis; Deep Transfer Learning for Crop Yield Prediction with Remote Sensing Data
ANSWER: Thank you for the comments and the references. In a nutshell, the most important question of interest in our study is the assessment of the different deep learning architectures as feature extractors in the context of multi-label classification of remote sensing images. While this has received substantial effort from the research community for simpler, multi-class classification tasks, it has not been studied for multi-label classification. We believe that land use/land cover classification from RSI should be treated as a multi-label classification task - the image patches typically do not contain a single, isolated object but rather multiple objects, which leads to multiple labels in terms of land-use/land-cover. Moreover, we investigate the influence of the amount of available labeled images on overall performance. Along these lines, we make a large experimental study to select the best deep NN architecture and use it to learn features that are then used by ensembles of trees with proven state-of-the-art performance for the task of MLC (Bogatinovski et al, 2022). We have also discussed the suggested references in the revised manuscript (line 48) and now provide a wider overview on the main topic under study (references [X], [Y], [Z]).
(Jasmin Bogatinovski, Ljupco Todorovski, Saso Dzeroski and Dragi Kocev. Comprehensive Comparative Study of Multi-Label Classification Methods, Expert Systems with Applications Volume 203, 1 October 2022, 117215, 2022, https://doi.org/10.1016/j.eswa.2022.117215)
In the conclusion section, considering limited label data, You say "it is better to use the ImageNet pre-training and only fine-tune the last layer for classification" (521). Do you have any experiments to show this? Have you done any experiments to show which and how many layers should be fine-tuned to get better results?
ANSWER: Figure 6 in the revised manuscript gives the results showing that it is better to use ImageNet pre-trained models and only fine-tune the last layer for classification for datasets with limited amounts of labeled data (such as Ankara, UCM). These conclusions are supported across the variety of experiments conducted in the study. The goal of the study was not to look for the optimal number of layers for fine-tuning, but rather assess the border cases - either use the large ImageNet models as they are provided or continue their training on the available data.
Random augmentation on none RGB images, especially when different channels are statistically dependent is arguable. You are discussing a lot about augmentation and also put a figure to show the augmentation outputs! Could you please clarify how did you deal with this issue when you are using Hyperspectral datasets?
ANSWER: We did not perform random augmentation, but rather we use different augmentation techniques in random order. We do not analyze hyperspectral images. We have two datasets that have hyperspectral variants (Ankara and BigEarthNet) but we use only the RGB channels in our experiments. Should one use augmentations for hyperspectral images, the same order of augmentations needs to be applied across the different channels (e.g., if horizontal flipping is selected as an augmentation technique, then it should be applied to all channels).
Your main message in this paper is about the different strategies to fine-tune a network for a new application (RS). It is expected to review relevant papers in the introduction section. I think you can improve the introductions section. Some papers mentioned such as attention mechanism, LSTM, ... not sure if relevant. Looks random.
ANSWER: Thank you for the remark. We have now included additional references [1, 2, 6, 11, 11, 12, 13, 14] to align our work with pre-existing related work.

Reviewer 2 Report
Paper's contribution: The authors present a comparative analysis of methods for the multi-label classification (MLC) of remotely sensed images. They consider different deep learning architectures which are used either directly for MLC according to end-to-end approach, or as feature extractors combined tree ensemble classifiers. The proposed methodologies are experimentally evaluated on six image data sets of RSI with different level of complexity, type of image sources, resolution, and cardinality.
Overall, the paper looks very interesting. Firstly, it is focused on a most significant topic of multi-class labelling of images with many diverse applications. Secondly, the authors provide a thorough experimental validation of the methods:
1) The datasets used are sufficiently large to support the derived conclusions, also aided by data augmentation approaches.
2) They investigated different aspects in the transfer learning paradigm, including: a) the influence of pretrained to the prior fine tuning of networks, b) comparison of the different network structures, c) comparisons between the tree-ensemble classifiers and d) the effect of labelled data involved in the learning.
3) They provide the results by means of statistical significance measures, which supports further the conclusions drawn.
4) The authors provide adequate discussions regarding the different scenarios considered.
Third, the manuscript is well written and organized for the background material and the experimental setup details.
Concluding, in view of the above remarks I deem that the work is complete enough, therefore suggesting its acceptance for publication.
I only have a minor remark for better readability. Specifically, in the feature-extractor/tree ensemble classifier scenario, how the class label ranking is obtained and compared to the ground-truth ranks?
Since the ensemble tree classifiers provide single class label estimates how rankings are computed?
Author Response
REVIEWER 2
Paper's contribution: The authors present a comparative analysis of methods for the multi-label classification (MLC) of remotely sensed images. They consider different deep learning architectures which are used either directly for MLC according to end-to-end approach, or as feature extractors combined tree ensemble classifiers. The proposed methodologies are experimentally evaluated on six image data sets of RSI with different level of complexity, type of image sources, resolution, and cardinality.
Overall, the paper looks very interesting. Firstly, it is focused on a most significant topic of multi-class labelling of images with many diverse applications. Secondly, the authors provide a thorough experimental validation of the methods:
1) The datasets used are sufficiently large to support the derived conclusions, also aided by data augmentation approaches.
2) They investigated different aspects in the transfer learning paradigm, including: a) the influence of pretrained to the prior fine tuning of networks, b) comparison of the different network structures, c) comparisons between the tree-ensemble classifiers and d) the effect of labelled data involved in the learning.
3) They provide the results by means of statistical significance measures, which supports further the conclusions drawn.
4) The authors provide adequate discussions regarding the different scenarios considered. Third, the manuscript is well written and organized for the background material and the experimental setup details. Concluding, in view of the above remarks I deem that the work is complete enough, therefore suggesting its acceptance for publication.
I only have a minor remark for better readability. Specifically, in the feature-extractor/tree ensemble classifier scenario, how the class label ranking is obtained and compared to the ground-truth ranks? Since the ensemble tree classifiers provide single class label estimates how rankings are computed?
ANSWER: We use tree ensembles for multi-label classification (MLC), i.e., random forests of trees for MLC (Kocev et al. 2013) and extremely randomized trees for MLC (Kocev et al. 2020). In particular, we use their implementation in the SciKit-learn [24] software package. The tree ensembles for MLC simultaneously predict the probability of each of the multiple class labels: The labels can then be ranked based on these predicted probabilities. This is reflected in lines 393-399 of the revised manuscript.
(Kocev et al. 2013) Tree ensembles for predicting structured outputs. D Kocev, C Vens, J Struyf, S Džeroski. Pattern Recognition 46 (3), 817-833
(Kocev et al. 2020) Ensembles of extremely randomized predictive clustering trees for predicting structured outputs. D Kocev, M Ceci, T Stepišnik. Machine Learning 109 (11), 2213-2241

Reviewer 3 Report
This paper explores the possibilities of a multi-label classification scheme for remote sensing images using two strategies and several CNN architectures over different datasets. An extensive analysis using 8 CNN architectures is presented. In addition, a comparative study between the use of a pre-trained or a fine-tunned strategy is shown. For the experiments, seven different datsets were used.
In my opinion, the work is well-written. However, the code is not public, which makes this work less attractive for other researchers. Therefore, I think that it deserves publication after addressing the following minor points:
In general, the work should be completed with a few more references.
Regarding plain writing, as I said, it is good in my opinion. However, I think the following points are potential mistakes to correct (maybe not, I am not a native English speaker):
- Page 2 - Lines 37-39: References to the use of RSI on: weather, climate change, land use and land cover changes... This paragraph could be improved including several with several references.
- Page 2 - line 42: "... category to an image." At the end of this phrase it might be interesting to include a reference to a recent survey on RSI classification.
- Page 2 - line 69: "... the transfer learning paradigm...". I suggest to include a survey on Transfer Learning for RSI there.
Subsection datasets: Include information about the spectral size on all the datasets.
- Page 5 - Line 191: "For S2, twelve channels and for S1, 2...". Please, acronyms (S2 and S1) must be defined in advance.
Subsection 2.2.2: One reference should be included for random forest and another for extremely randomized trees.
- Page 8 - Line 303: "... settings, we us a single...". Include letter e -> "we use a single"
- Table 1: Include definition of: "L", "Card", "Dens", ...
Author Response
REVIEWER 3
This paper explores the possibilities of a multi-label classification scheme for remote sensing images using two strategies and several CNN architectures over different datasets. An extensive analysis using 8 CNN architectures is presented. In addition, a comparative study between the use of a pre-trained or a fine-tunned strategy is shown. For the experiments, seven different datsets were used.
In my opinion, the work is well-written. However, the code is not public, which makes this work less attractive for other researchers.
ANSWER: The complete source code and the datasets used to execute the study are publicly and freely available at https://github.com/marjanstoimchev/RSMLC. This makes our work conform to high standards of reproducible research.
Therefore, I think that it deserves publication after addressing the following minor points:
In general, the work should be completed with a few more references.
Regarding plain writing, as I said, it is good in my opinion. However, I think the following points are potential mistakes to correct (maybe not, I am not a native English speaker):
- Page 2 - Lines 37-39: References to the use of RSI on: weather, climate change, land use and land cover changes... This paragraph could be improved including several with several references.
ANSWER: Thank you for your comment. We added references as suggested in the comment (references [1] and [2] in the revised manuscript).
- Page 2 - line 42: "... category to an image." At the end of this phrase it might be interesting to include a reference to a recent survey on RSI classification.
ANSWER: Thank you for your comment. We added references as suggested in the comment (references [6] and [11] in the revised manuscript).
- Page 2 - line 69: "... the transfer learning paradigm...". I suggest to include a survey on Transfer Learning for RSI there.
ANSWER: Thank you for your comment. We added references as suggested in the comment (references [11], [12], [13] and [14] in the revised manuscript).
Subsection datasets: Include information about the spectral size on all the datasets.
ANSWER: Information on the spectral size of all datasets was already included in Table 1 (they are all RGB images and have 3 channels, except for the Ankara dataset which is hyperspectral and has 119 channels). Considering that we only use the 3 RGB channels, we now removed the spectral size from the table. The caption of Table 1 has been now reformulated and makes this clear.
- Page 5 - Line 191: "For S2, twelve channels and for S1, 2...". Please, acronyms (S2 and S1) must be defined in advance.
ANSWER: S1 and S2 stand for Sentinel-1 and Sentinel-2. This has now been indicated in the text of the main paper. The text in the revised manuscript reads as follows: “There are 590326 such patches for which Sentinel-2/ S2 (and later also Sentinel-1/S1) images are available. For S2, twelve channels and for S1, two channels are available. We only use three of the S2 channels (RGB images).“ See lines 183-188 in the revised manuscript.
Subsection 2.2.2: One reference should be included for random forest and another for extremely randomized trees.
ANSWER: Thank you for this comment. The text has now been reformulated to include two references as suggested: scikit-learn [24] implementation of Random Forest tree ensembles for multi-label classification (MLC) [33] and Extra Tree ensembles for MLC [34].
- Page 8 - Line 303: "... settings, we us a single...". Include letter e -> "we use a single"
ANSWER: Corrected! Thanks for spotting this.
- Table 1: Include definition of: "L", "Card", "Dens", ...
ANSWER: Thank you for this comment, it was indeed necessary to explain the meaning of the abbreviations. The new version of the Table 1 caption now reads as follows:
Description of the used RS multi-label datasets. |L| denotes the number of possible labels, Card denotes label cardinality (i.e., average number of labels per image), Dens denotes label density (average proportion of images labeled with a given label), N is the number of images in the dataset, of which Ntrain are in the train and Ntest in the test datasets. w x h is the dimension of the images (in pixels).
